# Cardiac Rehabilitation for Fontan Circulation Patients: A Systematic Review, and Meta-Analysis

**DOI:** 10.3390/medicina60111817

**Published:** 2024-11-05

**Authors:** Luna van de Ven, Ana Clara Félix, Joana Suarez, Jorge Dias, Fátima F. Pinto, Sérgio Laranjo

**Affiliations:** 1Department of Paediatric Cardiology, Children’s Hospital, University Medical Center Utrecht, P.O. Box 85090, AB 3508 Utrecht, The Netherlands; 2Unidade Local de Saúde São José EPE, Pediatric Cardiology Department, Hospital de Santa Marta, Reference Center for Congenital Heart Diseases, Member of the European Reference Network for Rare, Low Prevalence Complex Diseases of the Heart (ERN GUARD-Heart), 1150-199 Lisbon, Portugal; 3Unidade Local de Saúde São José EPE, Physical and Medical Rehabilitation Department, Hospital de Santa Marta, 1169-024 Lisbon, Portugal; 4Clínica Universitária de Cardiologia Pediátrica, Centro Clínico Académico de Lisboa, 1169-024 Lisbon, Portugal; 5Comprehensive Health Research Center, NOVA Medical School, Faculdade de Ciências Médicas, NMS, FCM, Universidade NOVA de Lisboa, 1099-085 Lisbon, Portugal

**Keywords:** Fontan circulation, cardiac rehabilitation, exercise training, univentricular heart, aerobic exercise, systematic review, meta-analysis

## Abstract

*Background and Objectives*: Despite advances in the surgical management of patients with Fontan circulation, their exercise capacity and quality of life remain significantly impaired. Exercise-based cardiac rehabilitation (CR) offers promising improvements in these areas, but the implementation and adherence to these programmes are often inconsistent. This systematic review and meta-analysis aimed to evaluate the safety, efficacy, and optimal exercise modalities for Fontan patients. *Materials and Methods:* A systematic search of PubMed, Scopus, Web of Science, and Cochrane Library was conducted on 24 August 2023. Studies were screened and assessed for quality using the Cochrane RoB Tool 2 and STROBE checklist. Meta-analysis was performed using a continuous random-effects model to determine the effectiveness of various CR interventions, including aerobic exercise training (AET), resistance training, and inspiratory muscle training (IMT). *Results*: A total of 26 studies (7 RCTs, 19 cohorts) comprising 22 distinct cohorts were included, with a total sample size of 428 Fontan patients. The interventions ranged from 4 weeks to 24 months and included AET (18 studies), resistance training (11 studies), and IMT (6 studies). The meta-analysis revealed significant improvements in exercise capacity, with a pooled mean difference in peak VO_2_ of 1.947 (95% CI: 1.491 to 2.402, *p* < 0.001). Subgroup analyses showed that combined AET and resistance training had the most robust effect, with a mean difference of 2.11 (95% CI: 1.57 to 2.65, *p* < 0.001). Home-based interventions showed significant benefits, while supervised and hybrid interventions did not show statistically significant differences. Publication bias was identified, particularly in home-based interventions, where smaller studies demonstrated larger effect sizes, as confirmed by Egger’s test (Intercept = 2.417, 95% CI: 1.498 to 3.337, *p* = 0.001). However, no significant bias was detected in supervised or hybrid interventions, which displayed symmetrical distributions in funnel plots and non-significant Egger’s test results. *Conclusions:* CR appears to be an effective intervention for improving exercise capacity in Fontan patients, particularly when combining AET with resistance training. Home-based programmes offer promising results, though the potential for publication bias, especially in smaller studies, warrants cautious interpretation of these findings. Further research is needed to refine protocols, explore long-term outcomes, and determine the underlying mechanisms, particularly for patients with more severe clinical presentations. The low incidence of adverse events across the studies reinforces the safety of these interventions.

## 1. Introduction

Congenital heart defects affecting a single ventricle are characterised by the underdevelopment or absence of one of the heart’s ventricular chambers, necessitating complex surgical interventions. The three-stage Fontan procedure has become crucial in establishing functional circulation for these patients, effectively separating the pulmonary and systemic blood flows to allow survival with a single functional ventricle. Improved perioperative management, particularly the reduction of arrhythmias, has led to better survival rates and outcomes. It is estimated that the population of patients living with Fontan circulation will double over the next two decades due to these advancements [1,2,3]. However, despite a significant improvement in outcomes, the 30-year survival rate remains at approximately 85%, highlighting the continued need for optimising long-term care [3,4].

The Fontan procedure is typically completed in three stages. The first stage, known as the Norwood procedure, is performed shortly after birth to reconstruct the aorta and ensure sufficient blood flow to the body. The second stage, the Glenn procedure, is usually completed within the first year of life, which redirects blood from the upper body to the pulmonary arteries, bypassing the heart. Finally, the Fontan completion surgery, performed around 2 to 4 years of age, directs blood from the lower body to the lungs. The result is a circulation in which venous blood flows passively to the lungs without the assistance of a sub-pulmonary ventricle, leaving the single functional ventricle responsible for maintaining systemic circulation. While this staged approach improves survival, it creates a unique set of physiological challenges for patients.

The absence of a right ventricle in Fontan circulation fundamentally alters the body’s ability to maintain adequate blood flow during physical exertion. In a healthy heart, the right ventricle pumps deoxygenated blood to the lungs, while the left ventricle pumps oxygenated blood to the rest of the body. In Fontan patients, pulmonary circulation relies on passive flow driven by central venous pressure, which is insufficient to meet increased oxygen demands during exercise. This passive mechanism limits cardiac output, particularly under physical stress, resulting in reduced exercise tolerance, fatigue, and diminished peak oxygen uptake (peak VO_2_). These limitations, combined with muscle atrophy and deconditioning often associated with a sedentary lifestyle, compound the physical challenges faced by Fontan patients [5,6].

Compounding these issues, Fontan patients are at risk for various long-term complications that can further compromise their health. One of the most significant concerns is the progressive decline in ventricular function, particularly in the single ventricle responsible for systemic circulation, which may show signs of failure over time due to chronic pressure overload. This decline in function can manifest as reduced exercise tolerance, fluid retention, and an increased risk of heart failure. Arrhythmias are also common in this population, often arising from altered cardiac anatomy and increased pressure load on the heart, further limiting the capacity for physical activity and increasing the risk of hospitalisations or sudden cardiac events.

Another frequent complication is the development of Fontan-associated liver disease (FALD), driven by chronic hepatic congestion due to elevated venous pressure. Over time, this can lead to liver fibrosis or cirrhosis, further complicating the patient’s health and potentially exacerbating fatigue and exercise limitations. Additionally, thromboembolic events are a notable concern, as the sluggish venous blood flow inherent to Fontan circulation predisposes patients to clot formation, increasing the risk of strokes or pulmonary embolism. These systemic complications, combined with the inherent limitations of Fontan circulation, severely restrict exercise capacity and physical activity in these patients.

On average, Fontan patients have significantly lower peak VO_2_ compared to healthy individuals, often achieving only 60–70% of predicted values for their age. This reduced exercise capacity not only impacts their physical health but also diminishes psychosocial well-being, contributing to a lower quality of life. Many Fontan patients lead sedentary lives, further exacerbating muscle atrophy and deconditioning, which perpetuates a cycle of inactivity and declining functional capacity [5,6]. Breaking this cycle is crucial to improving long-term health outcomes in this population.

Recent studies have suggested that structured exercise-based cardiac rehabilitation (CR) programmes may offer a means to break this cycle, enhancing physical fitness, endurance, and quality of life for Fontan patients. Aerobic exercise training (AET), resistance training, and inspiratory muscle training (IMT) have emerged as promising modalities for improving cardiopulmonary function and increasing exercise tolerance. AET focuses on improving oxygen uptake and cardiopulmonary efficiency, making it particularly beneficial for Fontan patients whose pulmonary blood flow relies on passive mechanisms. Resistance training targets muscle mass improvement, which can enhance venous return to the heart and improve systemic circulation. IMT, on the other hand, aims to strengthen respiratory muscles, improving ventilation and reducing the impact of restrictive pulmonary dynamics on exercise capacity [7,8,9,10,11,12]. Together, these interventions seek to optimise residual cardiac function and counteract the deconditioning common in Fontan patients.

However, despite the growing body of evidence supporting the benefits of CR in Fontan patients, the practical implementation of these programmes remains limited, and adherence to prescribed exercise guidelines is often poor. The complexity of Fontan physiology, combined with the variation in patient health status, necessitates a tailored approach to exercise rehabilitation. Furthermore, there is no clear consensus on the optimal combination of AET, resistance training, and IMT, nor is there agreement on the best strategies for integrating these modalities into clinical practice. The diversity in study designs, patient populations, and exercise protocols adds to the challenge of determining best practices for CR in this population.

This systematic review and meta-analysis aims to provide a comprehensive evaluation of the current evidence regarding the safety, efficacy, and optimal modalities of exercise-based CR for Fontan patients. By critically appraising existing research, this study seeks to inform clinical practice and develop a structured approach to integrating exercise into the long-term management of Fontan patients. The goal is to improve exercise capacity, enhance quality of life, and address the complex physiological and haemodynamic challenges faced by this growing population of patients [1,2,3,4,5,6,7,8,9,10,11,12].

## 2. Materials and Methods

This systematic review and meta-analysis were conducted in accordance with the Preferred Reporting Items for Systematic Reviews and Meta-Analyses (PRISMA) guidelines. Although the review was not registered in a formal registry, a detailed protocol was prepared and is available upon request from the corresponding author.

### 2.1. Search Strategy

In collaboration with librarians from Nova Medical School, we devised a search strategy using terms and synonyms related to Fontan circulation and cardiac rehabilitation (CR) (detailed in the Appendix A). On 24 August 2023, we conducted a comprehensive search across PubMed, Scopus, Web of Science, and Cochrane Library for articles published between 1970 and August 2023 in English, French, Portuguese, and Spanish. The screening and evaluation of the 2005 identified studies were performed over an extended period. Any relevant articles published after that date were manually included in the search results.

Eligibility Criteria:

This systematic review examined the effects of CR in patients with Fontan circulation. To be included, studies had to meet the following criteria:

Population: Patients diagnosed with univentricular hearts and Fontan-type circulation.

Intervention: CR programs involving exercise training, lifestyle counselling, or medication management.

Outcomes: Reported effects of cardiac rehabilitation, such as mortality, quality of life, exercise tolerance, cardiac function, and hospital readmissions. Only primary research articles were considered, particularly Randomized Controlled Trials (RCTs).

Exclusion criteria included: (A) Studies lacking univentricular heart patients or those that did not separately analyse this group. (B) studies focused solely on surgical or pharmacological treatments without a CR component. (C) Studies without clear outcome measures for the efficacy of CR. (D) Articles unavailable in full text through PubMed, Scopus, Web of Science, or Cochrane Library. (E) Systematic reviews, meta-analyses, case reports, editorials, and commentaries were excluded.

Outcomes of Interest:

Primary outcomes:Exercise capacity, measured by peak VO_2_ and peak workload, which are central to assessing the physical capacity improvements through CR in Fontan patients.

Secondary outcomes:Cardiac biomarkers, including NT-proBNP, which serve as indicators of cardiac stress and heart function.Lung function, particularly changes in ventilatory efficiency and respiratory capacity.Lower-limb muscle strength, reflecting the effectiveness of resistance training components of CR.Quality of life, using validated scales or patient- and parent-reported measures, to assess the psychosocial impact of CR.Safety outcomes, including the incidence of adverse events during CR.

### 2.2. Data Extraction

Searches from PubMed, Scopus, Web of Science, and Cochrane Library were consolidated in Rayyan QCRI for deduplication and initial screening. Two researchers independently assessed articles, resolving discrepancies through consensus. Titles, abstracts, and full texts were screened against predefined criteria to exclude irrelevant studies. Manually identified relevant studies outside the initial search were included. Data extraction focused on study characteristics and the impact of cardiac rehabilitation on Fontan patients, and it was conducted from the final selection of articles.

### 2.3. Study Risk of Bias Assessment

The assessment of bias was conducted using two distinct tools based on the study design. For randomised controlled trials (RCTs), the Cochrane Collaboration’s Risk of Bias Tool 2 (RoB 2) was employed [13]. This tool evaluates five domains of potential bias, with each study classified as having low risk, some concerns, or high risk of bias. For cohort studies, the STROBE (Strengthening the Reporting of Observational Studies in Epidemiology) checklist was utilised [14]. This checklist comprises 22 items covering various elements of the study design. Studies were deemed to be sufficiently reported if they met over 70% of the criteria [15].

Two independent reviewers assessed the risk of bias for each study. Any discrepancies between the reviewers were resolved through discussion and consensus. In cases where consensus could not be achieved, a third reviewer was consulted. No automation tools were employed in the bias assessment process.

### 2.4. Effect Measures

For each outcome in this systematic review and meta-analysis, the effect measures were specified according to the nature of the data. Continuous outcomes, such as exercise capacity (measured by peak VO2 and peak workload), were synthesised using the mean difference (MD) with corresponding 95% confidence intervals (CIs). In cases where different scales were used across studies, the standardised mean difference (SMD) was employed to accommodate the variability in measurement units.

### 2.5. Synthesis Methods

#### Study Selection for Synthesis

The eligibility of studies for inclusion in each synthesis was determined by comparing the characteristics of the interventions against the predefined criteria outlined in the study protocol. Studies were initially tabulated, detailing intervention types, populations, outcomes, and methodologies. This tabulation allowed for comparison against the planned synthesis groups and ensured that only studies meeting the inclusion criteria were synthesised. Specifically, we focused on studies that reported on exercise interventions for patients with Fontan circulation, adhering to the predefined outcome measures.

### 2.6. Data Preparation

Prior to synthesis, the data were prepared by addressing missing summary statistics through imputation, where possible, using techniques such as calculating means from medians and ranges or estimating standard deviations from confidence intervals. For studies that required data conversion, such as standardising different units of measurement, appropriate transformations were applied to ensure comparability across studies.

### 2.7. Tabulation and Visual Display of Results

Results from individual studies were tabulated, presenting key information such as sample sizes, effect estimates, and confidence intervals. Visual representation of the data was achieved through the use of forest plots for meta-analyses, which clearly displayed effect sizes and heterogeneity. Additionally, funnel plots were utilised to assess potential publication bias.

### 2.8. Methods of Synthesis

The synthesis of results was performed using meta-analysis where appropriate, applying a random-effects model to account for expected clinical and methodological heterogeneity across studies. The choice of a random-effects model was justified by the variability in intervention types, populations, and outcomes. Heterogeneity was assessed using the I² statistic, with values over 50% indicating substantial heterogeneity. The entire analysis, including meta-analyses and the generation of forest plots, was conducted using IBM SPSS Statistics, version 29.0.2 (IBM Corp., Armonk, NY, USA).

### 2.9. Exploration of Heterogeneity

Subgroup analyses were conducted to explore potential sources of heterogeneity among study results. These analyses included stratification by intervention type (e.g., aerobic exercise, resistance training, inspiratory muscle training) and setting (e.g., home-based versus supervised rehabilitation). Where feasible, meta-regression was also considered to examine the impact of specific study characteristics on the outcomes.

### 2.10. Sensitivity Analyses

To assess the robustness of the synthesised results, sensitivity analyses were performed. These analyses involved excluding studies with a high risk of bias or studies with outlier results to determine whether the overall conclusions remained consistent. Additionally, alternative statistical models (e.g., fixed-effects models) were tested to evaluate the stability of the findings under different assumptions.

Reporting Bias Assessment

The potential for reporting bias due to missing results was considered throughout the synthesis process. As previously mentioned, funnel plots were generated for meta-analyses that included a sufficient number of studies to visually assess for asymmetry, which could indicate publication bias. Egger’s test was also utilised to statistically assess for small-study effects, which may suggest selective reporting. Additionally, the study selection process included a thorough comparison of reported outcomes with those outlined in study protocols (where available) to identify potential instances of selective outcome reporting. Sensitivity analyses, as described earlier, were also employed to assess the impact of any identified reporting bias on the overall conclusions.

Certainty Assessment

The certainty of the evidence for each outcome was assessed using the GRADE approach. This method evaluates several factors: risk of bias, inconsistency of results, indirectness of evidence, imprecision of effect estimates, and potential publication bias. Each factor contributed to rating the certainty of the evidence as high, moderate, low, or very low.

Registration and Protocol

This systematic review and meta-analysis were not registered in a formal registry. However, a review protocol was prepared and can be accessed upon request by contacting the corresponding author. No amendments were made to the original protocol during the review process.

## 3. Results

### 3.1. Systematic Review

#### 3.1.1. Study Selection

After removing duplicates, 1140 articles were identified, and six additional articles were found through a manual search. Of 44 articles screened in full, 26 [16,17,18,19,20,21,22,23,24,25,26,27,28,29,30,31,32,33,34,35,36,37,38,39,40,41] met the inclusion criteria. Figure 1 details the selection process and exclusion criteria.

#### 3.1.2. Study and Intervention Characteristics

Among the 26 studies (7 RCTs, 19 cohort studies), there were 22 distinct cohorts. Some studies examined the same cohorts while measuring different outcomes. Sample sizes ranged from 5 to 61 patients, with follow-ups lasting 4 weeks to 24 months. Interventions included:Aerobic Exercise Training (AET): 18 studiesResistance Training: 11 studiesInspiratory Muscle Training (IMT): 6 studies

Rehabilitation programs were either hospital-based [12] or home-based [14], with weekly frequencies ranging from daily to once per week. Program durations varied from 4 to 12 months (see Appendix A for details).

#### 3.1.3. Patient Characteristics

A total of 428 Fontan patients actively participated in CR programs across the included studies. The control groups were stratified into Fontan patients (*n* = 98) and healthy matched controls (*n* = 32). Ages ranged from 7 to 45 years, including both children and adults. Both genders were evenly represented, and patient characteristics like dominant ventricle and age at Fontan completion varied widely (see Appendix A).

#### 3.1.4. Quality Assessment

According to the Cochrane tool, all RCTs showed some concern of bias, while cohort studies met the STROBE checklist criteria (>70%). Appendix A provide more details.

### 3.2. Study Observations

The effects of CR on several observations are summarised in Appendix A.

#### 3.2.1. Exercise Capacity

Peak oxygen uptake (peak VO_2_) was the primary measure used to evaluate the impact of cardiac rehabilitation (CR) on patients with Fontan circulation. Most studies assessed peak VO_2_ using cardiopulmonary exercise testing, with the exception of Jacobsen et al. (2016), who utilised a shuttle run test. Across the 22 study cohorts reviewed [16,17,18,19,20,21,22,23,24,25,26,27,28,29,30,31,32,33,34,35,36,37,38,39,40,41], 11 reported significant improvements in peak VO_2_ following CR [17,18,20,25,26,27,29,34,35,36,39,40], while 12 found no notable changes [16,19,21,22,23,24,26,28,30,31,32,33,37,38,41].

Among the studies showing improvements, Turquetto et al. (2021) delivered particularly striking results [20]. This randomised controlled trial, involving 20 patients, recorded a 23% increase in peak VO_2_ after aerobic exercise training (AET), with values rising from 27.0 to 33.3 mL/kg/min (*p* = 0.012). Additionally, a 9% improvement was observed following inspiratory muscle training (IMT), with peak VO_2_ increasing from 26.6 to 29.1 mL/kg/min (*p* = 0.008). The intervention’s combination of supervised AET sessions with home-based IMT underscored the versatility of these training settings in improving oxygen uptake.

Other studies similarly demonstrated positive effects. Minamisawa (2001), a cohort study involving 11 patients, reported a significant improvement in peak VO_2_ following a home-based and supervised AET programme over 2–3 months [17]. Likewise, Opocher et al. (2005), working with a younger cohort of 10 patients, found notable improvements after an 8-month aerobic training programme [18]. These studies suggest that both home-based and supervised approaches can yield positive outcomes, irrespective of patient age.

For younger populations, Jacobsen (2016) found significant gains in peak VO_2_ after a 12-week programme combining aerobic and resistance training for 13 children aged 10 [25]. Dirks (2022) reported similar results in a cohort of 18 patients (mean age: 16.5 years) following a 10-month programme involving cycling-based AET and IMT [27]. These findings further reinforce the effectiveness of CR, particularly in home-based settings, for younger and adolescent Fontan patients.

Additional studies also reported positive outcomes. Perrone et al. (2022) demonstrated improvements in peak VO_2_ after three weekly AET sessions, each lasting 40 min [29]. Significant gains were also observed by Wittekind (2018) [34], Ait Ali (2018) [35], Cordina (2013) [36], and Scheffers (2023) [39], all of which involved various combinations of aerobic and resistance training. Even smaller studies like Bano (2023), with only five patients, reported improvements, suggesting that tailored CR programmes can be effective across diverse patient populations [40].

However, 12 studies did not observe significant changes in peak VO_2_ after CR. For instance, Neidenbach (2023), a randomised controlled trial with 20 patients, found no significant effects after a six-month IMT programme [16]. Similarly, Fritz (2020), who studied 42 patients over six months, did not detect improvements after IMT [19]. Other studies, such as those by Dulfer et al. (2014, 2015) [21,22,23], Sutherland (2018) [24], and Jacobsen (2018) [26], utilised a combination of aerobic and resistance training but saw no clear impact on oxygen uptake. Studies by Avitabile (2022) [28], Pyykkönen (2022) [30], Wu (2018) [31], Hedlund (2018) [32,33], Longmuir (2013) [7], and Brassard (2006) [38] similarly found no significant improvements in peak VO_2_, reflecting the variability in CR responses across different patient groups and interventions.

#### 3.2.2. Peak Workload

Peak workload, assessed through cardiopulmonary exercise testing, was another key indicator used to evaluate the effects of CR in Fontan patients. Among the 13 cohort studies reviewed [16,17,24,28,30,31,32,33,34,35,36,38,39,41], six found no significant changes [16,24,32,33,35,38,41], while seven reported improvements [17,28,30,31,34,36,39], especially in interventions combining aerobic and resistance training.

One notable example comes from Scheffers et al. (2023), who observed an 18% increase in peak workload after a 12-week high-weight leg resistance training programme, coupled with a high-protein diet, in 28 patients [39]. This combination of targeted resistance training and nutritional support underscores the benefits of a comprehensive approach, particularly for improving muscle strength and metabolic function in Fontan patients.

Similarly, Cordina et al. (2013), in a study involving six patients, found a 24-watt increase in peak workload after a 20-week high-intensity total body resistance training programme [36]. This intervention aimed to enhance lower limb muscle function, aligning with broader research that highlights the effectiveness of resistance training in boosting work capacity.

In contrast, Avitabile et al. (2022) observed a 9.35-watt increase in peak workload in 20 patients following a 24-week lower-extremity-focused exercise programme that combined supervised and home-based sessions [28]. Similarly, Pyykkönen et al. (2022) reported improvements in both peak VO_2_ and peak workload following a 6-month home-based programme involving aerobic and bodyweight resistance exercises [30].

Meanwhile, Wittekind et al. (2018) documented gains in peak workload among 10 patients who underwent a 12-week programme of aerobic exercise combined with low-resistance, high-repetition strength training [34]. The results suggest that even moderate resistance training, when integrated into aerobic programmes, can positively impact exercise capacity.

Conversely, six studies, including Neidenbach (2023) [16] and Wu (2018) [31], reported no significant changes in peak workload after IMT-focused interventions. Other studies by Minamisawa (2001) [17], Sutherland et al. (2018) [24], and Hedlund et al. (2018) [32,33] similarly found no significant changes, suggesting that the variability in CR responses may be influenced by factors such as exercise type, patient age, and the duration or intensity of the intervention.

Overall, these studies suggest that resistance training, either alone or in combination with aerobic training, appears most effective for improving work capacity in Fontan patients. Programmes focusing on lower limb strength and high-intensity resistance exercises reported the largest gains in peak workload, while those focused solely on aerobic or IMT interventions showed more limited success.

#### 3.2.3. VE/VCO2 Slope

The VE/VCO_2_ slope, a marker of ventilatory efficiency, was evaluated in 13 cohort studies. This measure is particularly relevant for Fontan patients as it reflects the relationship between ventilation and carbon dioxide output during exercise. While 10 studies found no significant changes [16,19,20,21,22,23,29,31,32,33,35,38,40], three documented improvements [30,34,41], suggesting that targeted training interventions can positively impact ventilatory efficiency.

One such study, Wittekind et al. (2018), involving 10 patients, found that a 12-week programme combining aerobic and low-resistance, high-repetition strength training significantly improved the VE/VCO_2_ slope [34]. Similarly, Pyykkönen et al. (2022) observed improvements in the VE/VCO_2_ slope after a 6-month home-based programme combining aerobic and bodyweight resistance exercises, highlighting the importance of multi-component interventions in enhancing ventilatory efficiency [30].

Additionally, Laohachai et al. (2017), which focused solely on IMT in 23 patients, reported significant improvements in VE/VCO_2_ slope following a 6-week home-based IMT programme [41]. This study emphasises the potential of respiratory muscle training to improve ventilatory efficiency even without broader aerobic or resistance training.

In contrast, the majority of studies, including Minamisawa (2001) [17], Neidenbach (2023) [16], and Sutherland (2018) [24], which typically employed aerobic training (AET), did not report improvements in ventilatory efficiency. This suggests that resistance training or respiratory-focused interventions, such as IMT, may be necessary to see meaningful changes in ventilatory efficiency in Fontan patients.

#### 3.2.4. Activity Levels

Activity levels were measured in seven studies [22,25,28,30,32,33,37,39], with four using accelerometers for objective tracking. The results were mixed. Duppen (2014, 2015) [22], Jacobsen (2016) [25], and Scheffers (2023) [39] found no significant increase in activity levels despite improvements in exercise capacity. These studies, focused primarily on aerobic exercise, suggest that enhanced exercise capacity does not necessarily translate into greater daily activity.

However, Longmuir et al. (2013) [37] reported a significant increase in moderate-to-vigorous physical activity in 30 children after a 12-month CR programme, with an average weekly increase of 36 ± 31 min at 24 months (*p* = 0.04). This suggests that long-term CR programmes, with sustained support and follow-up, may positively influence daily physical activity.

Self-reported measures, as used in Hedlund et al. (2018) [32,33] and Pyykkönen et al. (2022) [30], produced mixed results. Hedlund initially observed increased activity after a 12-week supervised AET programme [32], but these gains were not maintained at the 1-year follow-up [33]. In contrast, Pyykkönen reported stable improvements following a 6-month home-based programme [30], suggesting that home-based formats may foster greater adherence to daily physical activity.

#### 3.2.5. Cardiac Output

Cardiac output, a critical measure of heart function, was assessed in eight studies using MRI or echocardiography to evaluate the impact of cardiac rehabilitation (CR) on patients with Fontan circulation [21,22,23,28,31,34,36,39,40,41]. Among these, three studies [36,39,41] reported significant increases in cardiac output following CR, demonstrating the potential of targeted interventions to enhance heart performance in this patient population.

One notable finding came from Laohachai et al. (2017) [41], who focused on the effects of inspiratory muscle training (IMT) in a cohort of 23 patients. After a 6-week home-based IMT programme, resting cardiac output improved by 0.3 L/min (*p* = 0.03). This increase highlights the positive impact of respiratory-focused interventions like IMT on resting cardiac function in Fontan patients. Strengthening the respiratory muscles likely improves pulmonary efficiency and enhances heart-lung interactions, contributing to better cardiac performance at rest.

In another study, Scheffers et al. (2023) evaluated the effects of high-weight leg resistance training combined with a high-protein diet in 28 patients [39]. The study reported a significant increase in single-ventricle stroke volume, rising from 43 to 46 mL/beat/m² (*p* = 0.014). Stroke volume is a key determinant of cardiac output, and these findings suggest that resistance training can enhance the heart’s ability to pump blood more effectively with each beat. The focus on large muscle groups, such as the legs, likely contributed to improved venous return and overall cardiac function.

Similarly, Cordina et al. (2013) observed improvements in stroke volume both at rest and during exercise in a group of trained Fontan patients compared to a detraining period [36]. This study, involving 6 patients, found that resting stroke volume increased from 66.7 ± 9.3 mL during the detraining period to 77.8 ± 10.3 mL after the training programme (*p* = 0.01). Stroke volumes during exercise were also significantly higher. These results suggest that consistent training can improve both resting and dynamic cardiac output, further supporting the role of physical activity in maintaining and enhancing cardiovascular function in Fontan patients.

However, despite these positive outcomes, the majority of studies—five out of eight—did not report significant changes in cardiac output following CR [21,22,23,28,31,34,40]. These studies, which included a range of interventions from aerobic training to mixed aerobic and resistance programmes, suggest that improvements in cardiac function may not be consistently achievable across all patient groups or intervention types. The variability in results could be influenced by factors such as the baseline cardiac function of participants, the intensity and duration of the rehabilitation programmes, or the specific focus of the interventions.

#### 3.2.6. Cardiac Biomarkers

The effect of cardiac rehabilitation (CR) on cardiac biomarkers was evaluated in three cohort studies [23,29,39], with a focus on NT-proBNP, a biomarker commonly used to assess heart failure and cardiac strain. The results were mixed, with one study showing significant improvements while the other two found no notable changes.

Perrone et al. (2022) observed the most significant effect, reporting a substantial reduction in NT-proBNP levels following a 4-week aerobic exercise training (AET) programme [29]. Involving 12 patients, the study found that NT-proBNP levels decreased from 96.3 ± 6.7 pg/mL to 62.5 ± 46.1 pg/mL (*p* < 0.001), indicating a significant reduction in cardiac stress. This suggests that the AET regimen not only improved exercise capacity but also positively impacted underlying cardiac health by reducing strain on the heart.

By contrast, both Duppen et al. (2015) [23] and Scheffers et al. (2023) [39] reported no significant changes in cardiac biomarkers following their respective CR programmes. Duppen’s study, which included 26 patients and involved a combination of aerobic and resistance training, found no changes in NT-proBNP, despite improvements in other measures such as exercise capacity. Similarly, Scheffers’ study, which combined high-weight leg resistance training with a high-protein diet in 28 patients, did not observe any shifts in biomarkers, even though there were gains in stroke volume and muscle function.

#### 3.2.7. Lung Function

Lung function was assessed in nine studies [16,19,20,27,31,32,33,35,38,41], most of which incorporated inspiratory muscle training (IMT) as part of their cardiac rehabilitation (CR) programmes. Given the close interplay between pulmonary and cardiac function in Fontan patients, IMT, which strengthens the respiratory muscles, is particularly relevant for this group. While most studies reported improvements in lung function following CR, one study notably did not.

Out of the nine cohort studies, eight generally reported improvements in lung function [16,19,20,27,31,33,35,41]. Laohachai et al. (2017) [41], which involved 23 patients, found significant improvements in ventilatory efficiency and resting cardiac output after a 6-week IMT programme. Similarly, Wu et al. (2018), studying 11 patients, reported better pulmonary function after a 12-week home-based IMT intervention, highlighting the effectiveness of IMT in strengthening respiratory capacity [31]. Neidenbach et al. (2023) also found lung function improvements in a 6-month IMT programme with 20 patients, showing how longer interventions can further enhance pulmonary performance [16]. Dirks et al. (2020), who included both aerobic exercise and IMT in their 10-month programme for 18 patients, also observed improvements in lung function, underscoring the benefits of combining aerobic and respiratory training [27].

In contrast, Brassard et al. (2006), which focused on a combination of aerobic exercise training (AET) and resistance training without IMT, did not demonstrate a statistically significant improvement in lung function [38]. This study, involving five patients, suggests that AET and resistance training alone might not be sufficient to impact lung function significantly.

#### 3.2.8. Lower Limb Muscle Function

Lower limb muscle function was assessed in four studies that incorporated strength and resistance training as part of their cardiac rehabilitation (CR) programmes [28,30,38,39]. Improving lower limb strength is particularly important for Fontan patients, as it can enhance venous return and boost exercise capacity by reducing the strain on the cardiovascular system. However, the results from these studies were mixed, with only one reporting significant improvements in leg muscle strength [39].

Scheffers et al. (2023) [39] stood out due to its focus on high-weight resistance training combined with a high-protein diet. This study, involving 28 patients, was the only one to show significant improvements in lower limb muscle strength following CR. The combination of resistance training and nutritional support appeared to enhance muscle hypertrophy and strength, leading to notable gains in leg muscle function. The inclusion of a high-protein diet likely played a crucial role in supporting muscle recovery and growth, amplifying the effects of the high-intensity exercises. This study highlights the importance of combining both targeted exercise and dietary support to improve muscle function in this patient population.

In contrast, the other three studies—Avitabile (2022) [28], Pyykkönen (2022) [30], and Brassard (2006) [38]—did not observe significant improvements in lower limb muscle function, despite incorporating strength and resistance training. Unlike Scheffers’, these studies did not include a dietary component, which may explain the differing outcomes. Avitabile and Pyykkönen both focused on lower-extremity exercises, but neither reported notable gains in muscle strength after CR. Similarly, Brassard integrated resistance training into the rehabilitation programme but found no statistically significant improvements in leg muscle function.

The absence of significant improvements in these studies suggests that while resistance training alone may benefit overall exercise capacity, it might not be enough to produce measurable gains in muscle strength without additional nutritional support. Nutrition plays a critical role in muscle repair and growth, particularly for Fontan patients, who may have limited baseline muscle function. Therefore, the combination of resistance training and adequate protein intake may be necessary to fully optimise muscle performance in this group.

#### 3.2.9. Quality of Life

Quality of life (QoL) was evaluated in eleven studies focusing on Fontan patients undergoing cardiac rehabilitation [20,21,24,25,26,27,28,31,32,39,40]. Given the complex medical and psychosocial challenges faced by these patients, improvements in QoL are a crucial measure of CR’s effectiveness. Of the eleven studies reviewed, eight reported improvements in QoL following CR [20,21,24,25,26,32,39,40], while three found no significant changes [27,28,31].

Most studies—six out of eleven—assessed QoL through both patient and parent reports, providing a dual perspective on the impact of CR. A notable example is Jacobsen et al. (2016), which observed improvements in parent-reported QoL at both 12 weeks and 6 months post-intervention [25]. Parents reported significant gains in their children’s overall well-being and daily functioning, reflecting the perceived benefits of CR in supporting their health. However, the same study revealed a decline in patient-reported QoL at the 6-month follow-up. This contrast suggests that while parents may see improvements, patients themselves may experience new challenges, such as fatigue or the emotional burden of managing their condition during rehabilitation. This divergence between parent and patient perspectives highlights the complexity of assessing QoL in Fontan patients, where perceptions of improvement may vary depending on factors like age, understanding of the condition, and individual experiences of the rehabilitation process.

Other studies that reported improvements in QoL employed similar methods. For instance, Sutherland et al. (2018) [24] and Wu et al. (2018) [31] found that both patient and parent reports indicated better physical functioning and emotional well-being following CR. These interventions, which incorporated a combination of aerobic and resistance training, likely contributed to the physical and psychological improvements seen in the Fontan patients, supporting the idea that multi-faceted CR programmes can enhance both physical and emotional aspects of QoL.

On the other hand, three studies, including Dirks et al. (2022) [27], reported no significant changes in QoL post-CR. Dirks’ study, which combined cycling-based AET and IMT, demonstrated improvements in exercise capacity, but these physical gains did not translate into better QoL. This disconnect suggests that physical improvements alone may not be enough to impact perceived QoL, especially if psychosocial or emotional challenges remain unaddressed. The lack of significant QoL improvements in other studies may also be due to shorter rehabilitation programmes or specific training modalities that may not have sufficiently impacted the patients’ daily experiences or emotional well-being.

#### 3.2.10. Adverse Events

Of the studies reviewed, 17 specifically addressed adverse events related to cardiac rehabilitation (CR). Among these, 14 studies reported no adverse events [16,17,18,20,24,25,27,28,29,30,31,34,40,41] linked to the interventions, indicating that CR is generally well-tolerated in Fontan patients. The absence of adverse events across the majority of studies suggests that both aerobic and resistance training approaches, as well as inspiratory muscle training (IMT), are safe for this population.

However, three studies did report mild adverse events [19,36,39], which were generally non-serious and transient. These included instances of muscle soreness and mild fatigue, typically occurring after the initial phases of exercise interventions. None of the reported adverse events required significant medical intervention or resulted in the discontinuation of rehabilitation programmes.

Overall, the low incidence of adverse events across the studies supports the safety profile of CR in Fontan patients, with the majority of interventions being well-tolerated. The few reported cases of mild discomfort reinforce the importance of individualised rehabilitation plans to ensure that exercise intensity is appropriate for each patient’s condition, further minimising the risk of adverse effects.

### 3.3. Meta-Analysis

Our meta-analysis evaluated the differential impact of various CR interventions, including AET, resistance training, and IMT, on patients after the Fontan procedure. The analysis used a continuous random-effects model with the mean difference as the primary metric.

#### Effectiveness of Intervention Types

The effectiveness of CR interventions was assessed by evaluating improvement in exercise capacity, measured by peak VO2. The overall pooled effect indicated a significant improvement, with a mean difference of 1.947 (95% confidence interval [CI]: 1.491 to 2.402 to -; *p* < 0.001), suggesting that rehabilitation benefits this patient population.

### 3.4. Subgroup Analyses

Subgroup analyses were performed to assess the efficacy of the different interventions.

IMT: This subgroup did not demonstrate a significant effect, with a mean difference of 0.270 (95% CI: −2.05 to 2,6; *p* = 0.82).AET: AET alone approached statistical significance, with a mean difference of 1.44 (95% CI: −0.04 to 2.92; *p* = 0.06), suggesting a potential benefit of AET in improving outcomes.Combined AET and Resistance Training: The combination of AET and resistance training provided a significant mean difference of 2.11 (95% CI: 1.57 to 2.65; *p* < 0.001), indicating the robust effect of this hybrid intervention.Combined AET and IMT: The addition of IMT to AET also yielded a favourable outcome, with a mean difference of 1.69 (95% CI: 0.38 to 3.00; *p* = 0.012).Resistance Training Alone: Resistance training alone did not reach significance, with a mean difference of 3.22 (95% CI: −2.36 to 8.8; *p* = 0.259).

These findings suggest that, while individual interventions may offer some benefits, the combination of AET with either resistance training or IMT appears more effective for patients following the Fontan procedure (Figure 2).

Subgroup analyses regarding rehabilitation settings (home-based, supervised, or a hybrid of both) showed only a significant improvement in peak VO2 with home-based rehabilitation (Figure 3).

### 3.5. Heterogeneity and Consistency Across Studies

The consistency of the results across the included studies was evaluated using the I² statistic. In our meta-analysis, an I² value of 0% was observed (*p* = 0.591), indicating no evidence of heterogeneity and suggesting that the variations in the study estimates are compatible with what would be expected by random chance alone. This enhances the reliability of the overall effect estimates, as it implies that the observed treatment effects are consistent across different interventions and study populations within the included literature.

Subgroup analyses also explored heterogeneity by examining the setting of rehabilitation (home-based vs. supervised). Results indicated a significant improvement in peak VO2 with home-based rehabilitation, but no significant differences were found between supervised and hybrid interventions, further supporting the consistency of the results across various contexts.

### 3.6. Effect of Cardiac Rehabilitation on Peak Work

We further investigated the influence of various CR interventions on Peak Work capacity, expressed as standardised mean difference (SMD) to accommodate the different scales used in the studies.

The overall effect size for all interventions combined was modest and not statistically significant, with an SMD of 0.175 (95% CI: -0.051–0.400; *p* = 0.128), suggesting a small and uncertain effect of CR on peak work capacity (Figure 4).

### 3.7. Subgroup Analyses

The impact of different interventions on peak work yielded the following results:IMT: No significant change (SMD: 0.148; 95% CI: −0.246 to 0.543; *p* = 0.461).AET: No significant improvement (SMD: 0.090; 95% CI: −0.344 to 0.524; *p* = 0.684).Combined AET and Resistance Training: No significant effect (SMD: 0.152; 95% CI: −0.299 to 0.603; *p* = 0.509).Resistance Training Alone: A non-significant trend toward improvement (SMD: 0.487; 95% CI: −0.273 to 1.247; *p* = 0.209).

The I² statistic was 0% (*p* = 0.920), indicating no observed heterogeneity and consistent study results. The lack of statistically significant changes across interventions suggests that the type of CR program may not significantly affect peak work capacity in Fontan patients.

### 3.8. Sensitivity Analyses

To assess the robustness of the synthesised results, sensitivity analyses were conducted by excluding studies with a high risk of bias and those identified as outliers. The exclusion of these studies did not significantly alter the overall conclusions, indicating that the results were robust to variations in study quality and extreme data points.

Additionally, we tested the impact of using different statistical models (e.g., fixed-effects models) on the meta-analysis results. These alternative models yielded similar conclusions, further supporting the robustness of our findings.

### 3.9. Reporting Biases

Publication bias was evaluated using funnel plots and Egger’s regression-based test, stratified by intervention type and rehabilitation setting, for the outcomes VO_2_ and peak work.

For peak VO_2_, the funnel plot (Figure 5) exhibited asymmetry, particularly in studies with larger standard errors, suggesting potential publication bias. This was confirmed by Egger’s test, with a significant intercept in the overall analysis (Intercept = 2.117, 95% CI: 1.436 to 2.798, *p* < 0.001), indicating a high likelihood of publication bias across all intervention types. When stratified by intervention type, significant publication bias was observed in studies combining aerobic exercise training (AET) and resistance training (Intercept = 2.396, 95% CI: 1.230 to 3.561, *p* = 0.007), indicating an over-representation of smaller studies with larger effect sizes. In contrast, no significant publication bias was detected in studies evaluating AET alone (Intercept = 0.590, *p* = 0.850), Inspiratory Muscle Training (IMT) (Intercept = −2.183, *p* = 0.513), or the combination of AET and IMT (Intercept = 0.851, *p* = 0.768).

When stratified by rehabilitation setting for peak VO_2_, the funnel plot (Figure 6) showed asymmetry, especially in home-based and hybrid interventions. Egger’s test confirmed significant publication bias in home-based interventions (Intercept = 2.417, 95% CI: 1.498 to 3.337, *p* = 0.001), suggesting smaller studies with larger effect sizes may disproportionately contribute to the results. The overall analysis also indicated significant publication bias (Intercept = 2.117, 95% CI: 1.436 to 2.798, *p* < 0.001). However, no significant bias was detected in supervised interventions (Intercept = −1.190, *p* = 0.775) or hybrid interventions (Intercept = −1.883, *p* = 0.579), with symmetrical distributions in the funnel plot for these subgroups.

For peak work, both funnel plots (Figure 7) and Egger’s test revealed asymmetry, particularly in home-based and hybrid interventions, where smaller studies demonstrated larger effect sizes. This pattern suggests potential publication bias, especially in these intervention types, where smaller studies with extreme results may be over-represented. In contrast, studies involving supervised interventions appeared symmetrically distributed around the mean, indicating a lower likelihood of publication bias.

The results of Egger’s test for peak work supported the visual assessment from the funnel plot. The intercept for the overall analysis was not significant (Intercept = −0.072, 95% CI: −0.326 to 0.181, *p* = 0.528), indicating no strong evidence of publication bias across all studies. Similarly, no significant bias was detected for AET (Intercept = −0.087, *p* = 0.828), resistance training (Intercept = −74.420, *p* = 0.922), IMT (Intercept = −15.439, *p* = 0.599), or AET combined with resistance training (Intercept = −10.844, *p* = 0.935).

While asymmetry was observed in the funnel plots for certain intervention types, the non-significant Egger’s test results suggest that any potential bias is unlikely to have significantly influenced the overall findings. Nonetheless, the presence of visual asymmetry, particularly in home-based interventions, warrants caution when interpreting these results.

The analysis of publication bias across both peak VO_2_ and peak work suggests that home-based interventions are more prone to publication bias, with smaller studies showing larger effects. Supervised interventions, on the other hand, appear to be less affected by such bias, with symmetrical funnel plots and non-significant Egger’s test results. Hybrid interventions showed some asymmetry, but Egger’s test did not reveal significant evidence of bias. As such, while the overall findings remain robust, particular attention should be given to potential biases in smaller studies, especially in home-based settings, to ensure accurate interpretations of the impact of cardiac rehabilitation on peak VO_2_ and peak work.

## 4. Discussion

Fontan Procedure and Postoperative Challenges

The Fontan procedure represents a critical milestone in the treatment of single-ventricle congenital heart defects, significantly enhancing survival rates, with a 30-year survival rate approaching 85%. However, despite these advancements, Fontan patients continue to face significant postoperative challenges, particularly related to reduced exercise capacity and quality of life. Decreased peak oxygen uptake (peak VO_2_), muscle function limitations, and compromised cardiopulmonary efficiency are common, underscoring the need for effective interventions. Cardiac rehabilitation (CR) offers a pathway to improving these deficits, but optimal exercise modalities and adherence to these programmes remain unresolved. This systematic review and meta-analysis aimed to clarify the safety and efficacy of CR interventions for Fontan patients and provide evidence-based recommendations for postoperative management.

Safety

Across the studies reviewed, CR was consistently shown to be safe, with 14 of the 17 studies explicitly reporting no adverse events linked to interventions. Mild and transient adverse events, such as muscle soreness and fatigue, were reported in only three studies. These events did not lead to the discontinuation of CR programmes, suggesting that the interventions were well-tolerated. However, it is important to note that many studies excluded Fontan patients with more severe clinical conditions, such as those with heart failure or significant arrhythmias. This exclusion presents a challenge in determining the safety of CR for patients who might benefit the most from exercise-based interventions. Including this high-risk population in future studies will provide a clearer understanding of the safety and efficacy of CR across the full spectrum of Fontan patients.

Impact of Exercise Training on Fontan Patients

Cardiac rehabilitation (CR) interventions have shown promising results in improving exercise capacity, muscle strength, and lung function in Fontan patients. The meta-analysis demonstrated a significant improvement in peak VO_2_, with a mean difference of 1.947 mL/kg/min (95% confidence interval [CI]: 1.491 to 2.402; *p* < 0.001), underscoring the effectiveness of CR in this population. Aerobic exercise training (AET) was particularly effective, with studies such as Turquetto et al. (2021) reporting a 23% increase in peak VO_2_ following AET, highlighting the potential of endurance training to enhance oxygen uptake.

The combination of AET with resistance training or inspiratory muscle training (IMT) appeared to yield even greater improvements in exercise capacity. For instance, Scheffers et al. (2023) demonstrated that high-weight resistance training, combined with a high-protein diet, resulted in an 18% increase in peak workload and significant muscle hypertrophy. The inclusion of a high-protein diet was crucial for enhancing the outcomes of resistance training, suggesting that future CR programmes could benefit from integrating dietary support alongside physical training.

In addition to gains in exercise capacity, cardiac biomarkers such as NT-proBNP responded positively to exercise training. For example, Perrone et al. (2022) reported a significant reduction in NT-proBNP levels, with values dropping from 96.3 ± 6.7 pg/mL to 62.5 ± 46.1 pg/mL (*p* < 0.001), suggesting decreased cardiac stress and potentially improved heart function. However, inconsistencies in biomarker outcomes across studies indicate that further investigation is needed to better understand the relationship between exercise training and long-term cardiac health in this population.

Lung function and lower-limb muscle strength were also positively affected by CR. High-weight resistance training, particularly when combined with a high-protein diet, resulted in considerable gains in muscle strength. While some studies, such as Brassard et al. (2006), did not report significant improvements in lung function, others, like Laohachai et al. (2017) and Wu et al. (2018), found notable enhancements following IMT, emphasising the importance of structured training programmes for addressing these deficits.

The effects of exercise training on quality of life (QoL) were more varied. Some studies, such as Jacobsen et al. (2016), found improvements in parent-reported QoL, while patient-reported QoL showed declines at longer follow-ups, reflecting the complex and multifaceted nature of quality of life in Fontan patients. Other studies, such as Sutherland et al. (2018) and Wu et al. (2018), also observed improvements in QoL, particularly in physical and emotional well-being. However, Dirks et al. (2022) found no significant changes in QoL despite improvements in exercise capacity. These discrepancies suggest that physical improvements alone may not always translate to enhanced QoL and underscore the need for a holistic approach that addresses both the physical and psychosocial dimensions of rehabilitation.

Efficacy of Different Cardiac Rehabilitation Interventions

The meta-analysis provided valuable insights into the efficacy of various CR interventions. Aerobic training alone approached statistical significance in improving exercise capacity (mean difference: 1.44, 95% CI: −0.04 to 2.92; *p* = 0.06), but combining AET with resistance training yielded the most substantial benefits (mean difference: 2.11, 95% CI: 1.57 to 2.65; *p* < 0.001). This combination demonstrated a synergistic effect, suggesting that addressing multiple physiological components of the Fontan circulation is more effective than single-modality interventions. Similarly, while IMT alone did not show significant effects, its combination with AET led to improved outcomes (mean difference: 1.69, 95% CI: 0.38 to 3.00; *p* = 0.012), underscoring the importance of multi-faceted exercise approaches.

The findings suggest that integrating multiple training modalities—such as endurance, strength, and respiratory exercises—delivers superior outcomes for Fontan patients, addressing both cardiovascular and musculoskeletal limitations.

Synergetic effect of different training modalities

As previously stated, the meta-analysis found that combining different training modalities yields better outcomes due to a synergistic effect. The reduced exercise capacity and cardiac health in Fontan patients result from complex interactions among various factors. Addressing the problem at multiple levels logically leads to more significant improvements.

The Fontan circulation lacks a sub-pulmonary pump, causing venous blood to drain directly into the lungs and limiting hemodynamics. This restricts cardiac augmentation during exercise and reduces exercise capacity [7,8]. As a result, Fontan patients often lead a sedentary lifestyle, worsening their muscle atrophy [5] and further reducing exercise capacity [42], creating a downward spiral. Tackling this issue on multiple fronts is essential.

AET can improve overall cardiorespiratory capacity, enabling more physical activity [20,24]. Resistance training helps increase muscle mass, making activity easier while potentially enhancing non-pulsative venous flow. This, in turn, boosts systemic preload, cardiac output, and exercise capacity [28,36,43]. IMT can also improve pulmonary blood flow by strengthening the thoracic pump [27].

Therefore, combining different training modalities targets multiple aspects of the Fontan circulation, potentially leading to a synergistic effect that produces the most favourable outcomes for these patients.

Heterogeneity and Bias in Study Findings

While the meta-analysis demonstrated clear improvements in peak VO_2_, several limitations need to be addressed. There was significant heterogeneity among studies in terms of patient characteristics, intervention designs, and outcome measures. Egger’s test revealed potential publication bias, particularly in home-based interventions, where smaller studies with larger effect sizes were over-represented (Intercept = 2.417, *p* = 0.001). This suggests that some of the more positive findings may be inflated by the disproportionate inclusion of smaller studies with extreme results. However, supervised and hybrid interventions did not show significant bias, and their symmetrical distribution in funnel plots adds to the reliability of their findings.

Addressing these limitations in future research, particularly through larger studies with robust study designs, will help mitigate the potential for bias and enhance the generalisability of the findings.

Limitations and Future Perspectives

Despite encouraging results, several limitations of the current research must be acknowledged. Most of the included studies had small sample sizes, and only seven were randomised controlled trials (RCTs). Furthermore, the exclusion of Fontan patients with more severe clinical conditions limits the generalisability of these findings to the broader Fontan population. The long-term effects of CR also remain uncertain, as few studies included follow-up periods beyond 12 months.

Future studies should focus on enrolling larger, more diverse cohorts, including patients with more complex clinical conditions. Multi-centre trials with extended follow-up periods are essential to better understand the sustainability of CR benefits. Investigating the mechanisms underlying improvements in exercise capacity, cardiac biomarkers, and quality of life will also be critical for refining CR protocols and optimising them for long-term use.

## 5. Conclusions

This systematic review and meta-analysis confirm that cardiac rehabilitation is a safe and effective intervention for improving exercise capacity and overall health outcomes in Fontan patients. The combination of AET with resistance training or IMT appears to provide the most significant benefits. However, further research is needed to extend these benefits to patients with more severe clinical conditions and to optimise CR protocols for long-term efficacy. The presence of publication bias in some studies highlights the need for caution in interpreting the more extreme positive findings. Nonetheless, CR remains a promising strategy for enhancing the quality of life and long-term health of Fontan patients, particularly when multi-modal interventions are employed to address the complex physiological challenges they face.

## Figures and Tables

**Figure 1 medicina-60-01817-f001:**
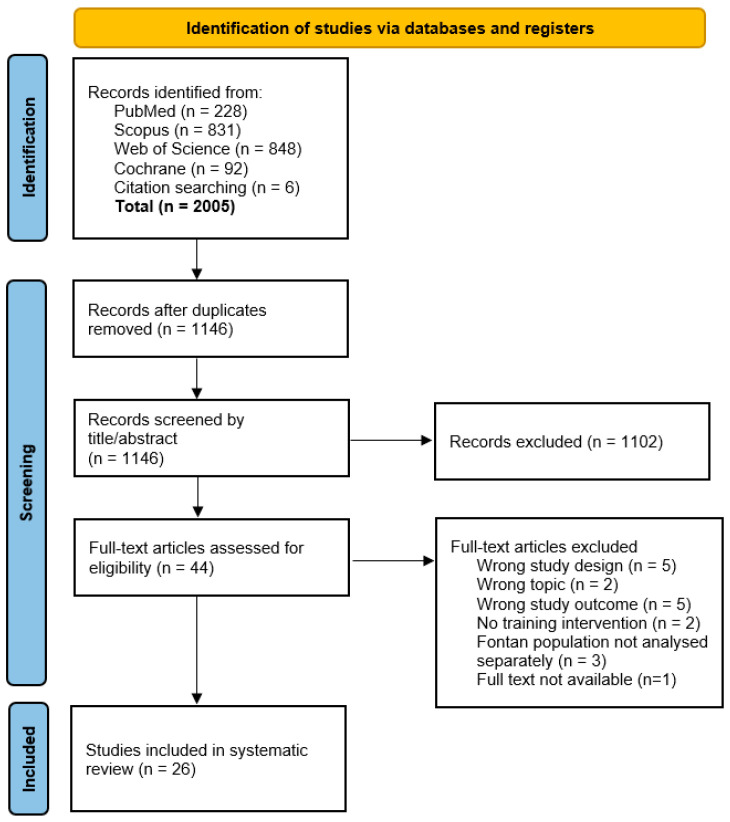
Prisma flow diagram of the study selection. In total, 1999 articles were identified by the online databases and 6 articles were manually included. After removing the duplicates, 1146 articles were screened. After screening by title/abstract, a full-text screening was done for 44 of the 1146 articles. Of these articles, 26 adhered to all inclusion criteria and were subsequently included in the systematic review and meta-analysis.

**Figure 2 medicina-60-01817-f002:**
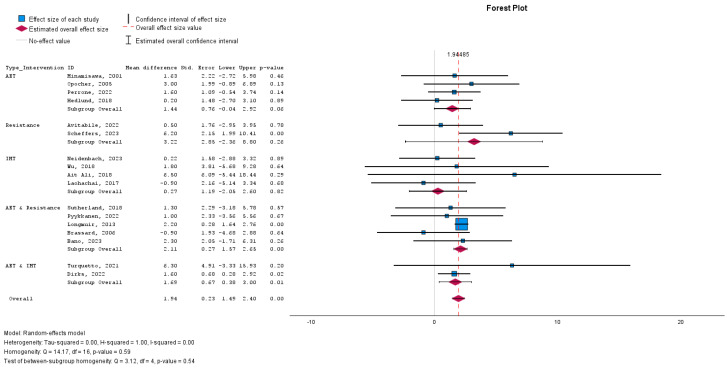
Forest plot demonstrating the effect of cardiac rehabilitation interventions by evaluating the parameter peak VO2, regarding intervention type, using a continuous random-effects model. The overall pooled effect showed a significant improvement in peak VO2 after participating in a cardiac rehabilitation intervention (*p* < 0.001). Subgroup analysis investigated the effect of different types of cardiac rehabilitation interventions. AET: Aerobic exercise training. IMT: Inspiratory muscle training.

**Figure 3 medicina-60-01817-f003:**
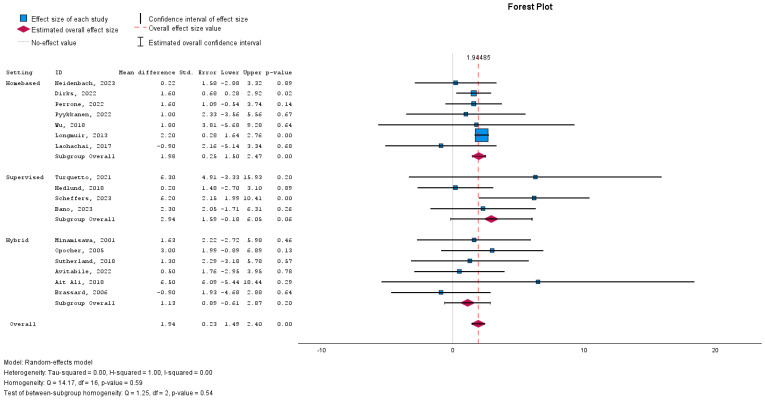
Forest plot demonstrating the effect of cardiac rehabilitation interventions by evaluating the parameter peak VO2, regarding rehabilitation setting, using a continuous random-effects model. The overall pooled effect showed a significant improvement in peak VO_2_ after participating in a cardiac rehabilitation intervention (*p* < 0.001). Subgroup analysis investigated if the setting of the cardiac rehabilitation intervention influenced the effect.

**Figure 4 medicina-60-01817-f004:**
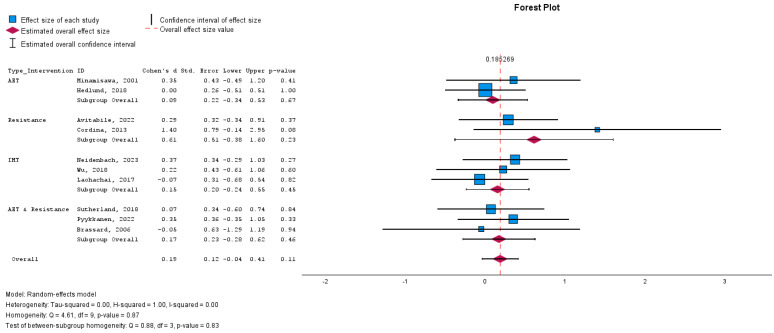
Forest plot demonstrating the effect of cardiac rehabilitation interventions by evaluating the parameter peak work, regarding intervention type, using a continuous random-effects model. The overall pooled effect did not show a significant improvement in peak work after participating in a cardiac rehabilitation intervention (*p* = 0.0128). Subgroup analysis investigated the effect of different types of cardiac rehabilitation interventions.

**Figure 5 medicina-60-01817-f005:**
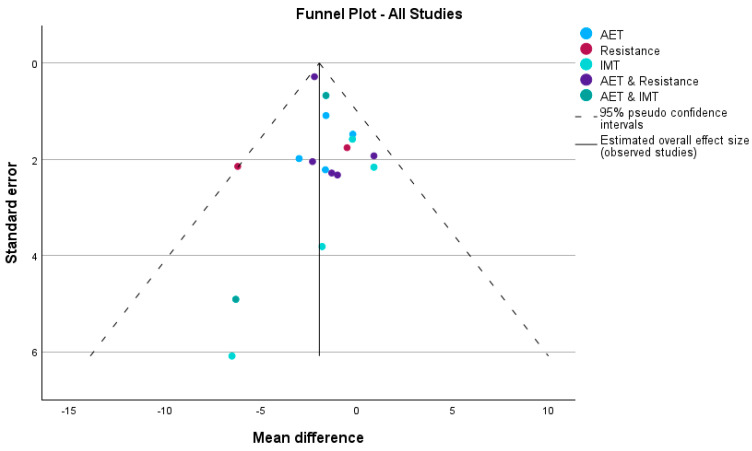
Funnel plot showing potential publication bias for peak VO_2_ across various intervention types. The plot includes aerobic exercise training (AET), resistance training, inspiratory muscle training (IMT), AET combined with resistance, and AET combined with IMT. The dashed lines represent the 95% pseudo-confidence intervals, and the solid line indicates the estimated overall effect size (mean difference) across all studies.

**Figure 6 medicina-60-01817-f006:**
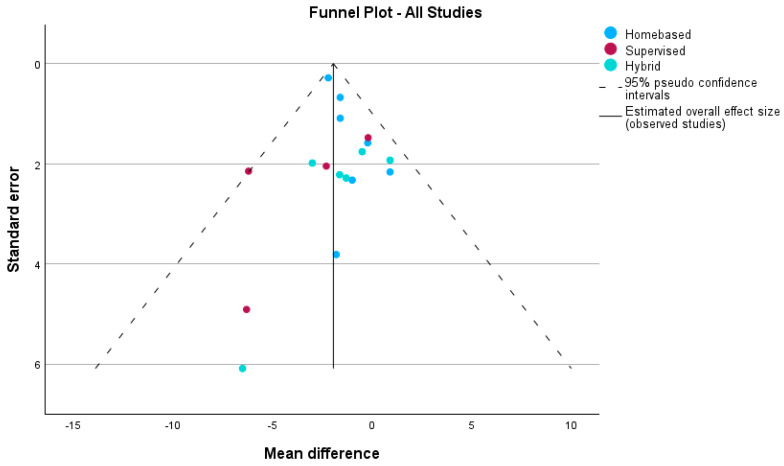
Funnel plot depicting potential publication bias in studies assessing peak VO_2_, stratified by rehabilitation setting: home-based, supervised, and hybrid interventions. The dashed lines represent the 95% pseudo-confidence intervals, while the solid line indicates the estimated overall effect size across all studies.

**Figure 7 medicina-60-01817-f007:**
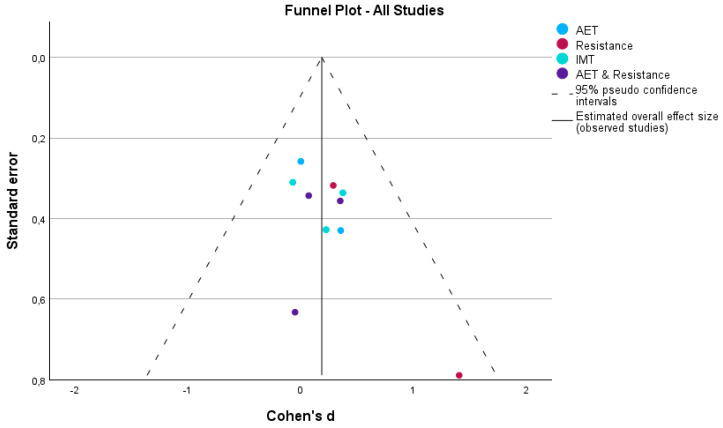
Funnel plot illustrating potential publication bias for peak work across all intervention types, including aerobic exercise training (AET), resistance training, inspiratory muscle training (IMT), and a combination of AET and resistance. The dashed lines represent the 95% pseudo-confidence intervals, while the solid line shows the estimated overall effect size (Cohen’s d).

## Data Availability

Data supporting the findings of this systematic review and meta-analysis are based on previously published studies, which are available in the public domain or through institutional access. No new data were generated or collected specifically for this study. The datasets and analyses used in this review can be accessed through the original publications referenced in the article. Template data collection forms that were used during the review process are available upon request by contacting the corresponding author. Additionally, any other materials, such as analytic scripts and tools used for data extraction and synthesis, can also be provided upon request.

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
