# Peer review of "Cardiac Rehabilitation for Fontan Circulation Patients: A Systematic Review, and Meta-Analysis"

_medicina, 2024, doi:10.3390/medicina60111817_

Round 1
Reviewer 1 Report
Comments and Suggestions for Authors
The authors performed a systematic review with meta-analysis examining the effect of cardiac rehabilitation for Fontan circulation patients. Their analysis showed improvement in peak VO2 but no change in peak work. They performed some subgroup analysis to further explore the effectiveness of CR. Overall, the subject is important but the manuscript must improved in several aspects.
1- The important part of the manuscript is the performed meta-analysis but in the results section of the abstract none of the numbers regarding the meta-analysis are mentioned. The authors are just interpreting the results instead of bringing the results of their meta-analysis.
2- The methods section of the abstract: It is unnecessary to talk about the quality assessment and screening application here. Instead, you should bring important data regarding how you performed the meta-analysis and what were your endpoints and other details.
3- The authors mention the funnel plots for assessing the risk of publication bias. The plots should be brought in the supplement. Also, bring the results of Egger's test rather than just saying it was not significant.
4- What was the minimum number of studies involved in an analysis to assess the funnel plot results? Also, when did the authors use a random-effects model instead of a fixed-effects model? The authors tend to provide some vague wording such as "when applicable" for these analyses.
5- For risk of bias assessment of RCTs using the RoB2 tool, the guideline recommends rating a study as "some concerns" if "The study is judged to raise some concerns in at least one domain for this result, but not to be at high risk of bias for any domain." According to this, the risk of bias for all the included RCTs should be "some concerns" instead of "low risk". The authors need to fix this both in their supplement and the text.
6- In the results section for both the systematic review and the meta-analysis, the authors need to cite all the mentioned studies discussed in the text.
7- Page 7, Section 3.3.1 Effectiveness of Intervention Types: The pooled MD comparing the group undergoing CR with the control group is -1.947 implying the peak VO2 (or the improvement in VO2) is lower in the intervention group and hence, the control group showed better results. There may be an error in performing this analysis. The authors may have reversed the intervention with the controls. Also, for performing this analysis, have they used the change in VO2 after the intervention or they have used the follow-up results? The forest plots have this limitation as they do not show what initial data are entered into the application. It may be better to re-perform the analysis with better applications such as RevMan.
8- The interpretation of the results of the meta-analysis should only be brought up in the discussion and not the results section (e.g. Line 343-345, 379-382, ...)
9- The outcomes of interest are not clearly defined. The authors should clearly mention their primary and secondary endpoints. Also, no dichotomous endpoint is analyzed, so why do the authors mention lines 127-129?
Author Response
Thank you for your insightful feedback. We have carefully addressed each of your points and made the necessary revisions to improve the manuscript. Below are our detailed responses:
1. Meta-analysis results in the abstract: We agree with your comment and have revised the abstract to include specific results from the meta-analysis. The updated abstract now clearly states the numerical results, such as the pooled mean difference in peak VOâ‚‚ of 1.947 mL/kg/min (95% CI: 1.491 to 2.402, p < 0.001), and highlights the results of the subgroup analyses, which showed the most substantial improvement with combined AET and resistance training (mean difference of 2.11, p < 0.001). These changes make the abstract more quantitative and aligned with the meta-analysis.
2. Abstract methods section: We have simplified the methods section of the abstract by focusing on the core aspects of the meta-analysis rather than the quality assessment and screening process. We now clearly describe the meta-analysis model used (random-effects) and specify that the primary endpoint was peak VOâ‚‚. The assessment of study quality has been moved to the methodology section of the main text.
3. Funnel plots and Egger’s test: As requested, we have added the funnel plots and the full results of Egger’s test in the supplementary materials. In the main text, we now provide precise values for Egger’s test, including examples such as Intercept = 2.417, p = 0.001, confirming publication bias in home-based interventions. Additionally, we have made it clear that no significant bias was found in supervised or hybrid interventions. These changes are reflected in both the results and the discussion.
4. Random-effects vs fixed-effects models: We have clarified the choice of random-effects models in the meta-analysis. As detailed in the methods, the random-effects model was used due to the expected clinical and methodological heterogeneity across studies. We also performed sensitivity analyses using fixed-effects models to ensure the robustness of the findings. This approach, along with our heterogeneity assessment (I² statistic), is now more clearly explained in the manuscript.
5. Risk of bias assessment: We appreciate your feedback on the use of the RoB2 tool. We have corrected the risk of bias classification for all randomised controlled trials (RCTs) to "some concerns" where appropriate. This change is reflected both in the supplementary material and in the main text. We have also discussed the implications of these assessments in the context of the study findings.
6. Citing all mentioned studies: We have reviewed the results section and ensured that all studies discussed are properly cited. All 26 studies are now clearly referenced, with detailed descriptions of their contributions to the overall findings.
7. Pooled mean difference interpretation: We thank you for pointing out the error in the pooled mean difference for peak VOâ‚‚. Upon review, we realised that the intervention and control groups had been inverted in the original analysis. We have corrected this and re-analysed the data using SPSS 29, ensuring that the results accurately reflect improvements in peak VOâ‚‚ in the intervention group. The corrected mean difference is now 1.947 mL/kg/min, favouring the intervention group.
8. Interpretation of meta-analysis results: We agree that the interpretation of the meta-analysis results should be confined to the discussion. We have revised the results section to focus solely on reporting the data, and we moved any interpretation or contextualisation of the findings to the discussion, where they are now properly framed.
9. Definition of outcomes: We have removed the mention of dichotomous endpoints and have now clearly defined the primary outcomes as improvements in peak VOâ‚‚ and peak workload. The secondary outcomes include cardiac biomarkers (e.g., NT-proBNP), lung function, and lower-limb muscle strength. This clarification has been integrated into both the methods and results sections, ensuring transparency about the aims of the analysis.
Conclusion:
We are grateful for your valuable feedback, which has greatly improved the quality and clarity of our manuscript. The changes we have made reflect your suggestions, particularly in making the meta-analysis results more prominent and ensuring clarity in the methodology and presentation of data. We are confident that these revisions address the issues raised and enhance the robustness of our findings.
Please refer to the revised manuscript and supplementary materials for the updated content.
Thank you for your time and consideration.
Reviewer 2 Report
Comments and Suggestions for Authors
Authors have submitted a quite interesting manuscript regarding the cardiac rehabilitation for fontan circulation patients. Despite the use of graphs it would be helpful for the reader to see some illustrations too. The citations used are up to date and give reader a current prospective regarding the topic. Despite the fact that the discussion section includes the limitations of the study and current prospective, it would be favorable to expand the conclusions section in order to give a more clear key message at the end.
Author Response
Thank you for your valuable feedback. We understand the suggestion to include illustrations, but after careful consideration, we decided not to include additional illustrations in the current manuscript. The primary reason is that the detailed graphs and funnel plots already provide a clear and comprehensive visual representation of the meta-analyses and the effect of cardiac rehabilitation (CR) interventions on Fontan patients. These visualisations effectively communicate the study's findings and help readers interpret the data without requiring further illustrative diagrams.
As for the Conclusions section, we aimed to provide a concise and focused summary of the study's key findings, as highlighted in the text:
- Safety and Effectiveness: We reaffirm that CR interventions are both safe and effective for improving exercise capacity and overall health in Fontan patients.
- Combination Therapy: The data suggest that combining aerobic exercise training (AET) with resistance training or inspiratory muscle training (IMT) yields the most significant benefits.
- Need for Further Research: While the current evidence is promising, more research is needed to optimise CR protocols, particularly for patients with more severe conditions.
- Publication Bias Caution: We emphasised the need for caution in interpreting more extreme findings, due to the publication bias identified in some studies.
This expanded conclusion balances the interpretation of the results with practical recommendations, while also acknowledging the study's limitations. It provides a clear and actionable message for clinicians and researchers alike. We believe that this approach offers clarity without overwhelming readers with unnecessary visual elements.
We appreciate your understanding and hope that this revised explanation and justification meet your expectations.
Reviewer 3 Report
Comments and Suggestions for Authors
Dear editor;
I reviewed the article entitled “Cardiac Rehabilitation for Fontan Circulation Patients: a Systematic Review, and Meta-Analysis” . My comments are below:
1- Authors stated that “On August 24, 2023, we conducted a comprehensive search across PubMed, Scopus, Web of Science, and Cochrane Library, filtering for articles in English, French, Portuguese, and Spanish published between 1970 and August 2023. “
2005 studies were evaluated. All these articles were searched in an only 1 day?? It is too hard.
2- Abstract, results section has been written as methods section. What is the results of the study. It should be mentioned in here. Abstract-results section should be re-written. The results of the studies should be added to this section.
3- there are many minor English typos in the article. English language proofediting is required.
4- The physiology of the fontan circulation should be explained in detail in the introduction. What is the fontan procedure. It should be defined for the readers. Also, it should be useful to insert an image of fontan Circulation to inform the reader about the subject.
5- Aerobic Exercise Training (AET), Resistance Training, Inspiratory Muscle Training (IMT) should be described in the methods section comprehensively.
6- A table may be added to the study, in which a brief information about included studies may be provided (study caharacteristics: year, design, patients, intervention, outcome, adverse events etc…)
Comments on the Quality of English Languagethere are many minor English typos in the article. English language proofediting is required.
Author Response
Thank you for your thorough review of our manuscript, “Cardiac Rehabilitation for Fontan Circulation Patients: a Systematic Review, and Meta-Analysis.” We appreciate your valuable feedback, which has helped us clarify and improve several aspects of our work. Below are our detailed responses to each of your comments:
1. Search process clarification: We understand the confusion regarding the timeline for the search process. To clarify, the search was indeed conducted on August 24, 2023. However, this refers to the date on which the comprehensive search was executed, not the completion of the article selection and evaluation. The screening, analysis, and evaluation of the 2005 identified studies were done subsequently over a longer period, ensuring a thorough and methodical approach to inclusion. We have adjusted the phrasing in the manuscript to make this process clearer.
2. Abstract results section: We agree with your point regarding the abstract. We have revised the results section of the abstract to clearly present the outcomes of the study, including the specific results from our meta-analysis, such as the pooled mean difference in peak VOâ‚‚, and the efficacy of different cardiac rehabilitation interventions. This change ensures that the results section now directly reflects the findings of our study rather than reading like a methods description.
3. English language proofreading: The manuscript has undergone a thorough English language review and proofreading by a native English speaker. We have addressed all minor typographical errors and inconsistencies to ensure the text is polished and clear.
4. Explanation of Fontan physiology and procedure: We have expanded the introduction to include a detailed explanation of the Fontan procedure, its physiological implications, and its impact on exercise capacity and long-term outcomes. The revised introduction now provides the necessary background for readers who may not be familiar with Fontan circulation. Regarding your suggestion to include an illustration of the Fontan circulation, we have opted to focus on textual explanation as this is a review and meta-analysis. However, we understand the value of visual aids and have made the information more comprehensive in the introduction to accommodate readers who may require more context.
5. Comprehensive description of AET, Resistance Training, and IMT: We have expanded the methods section to provide a more comprehensive description of the different rehabilitation modalities included in the studies. This includes a clear explanation of aerobic exercise training (AET), resistance training, and inspiratory muscle training (IMT), detailing their goals, mechanisms, and how they contribute to improving the health outcomes of Fontan patients. This additional detail strengthens the methodological clarity and allows readers to fully understand the scope of the interventions being evaluated.
6. Table of study characteristics: As per your suggestion, a detailed table summarising the study characteristics (including year, design, patient cohort, intervention types, outcomes, and adverse events) has been included in the supplementary material. This table provides a clear overview of the included studies and facilitates easier reference for readers.
We hope these revisions adequately address your concerns and enhance the quality and clarity of our manuscript. We sincerely appreciate your feedback, which has helped improve the structure and content of our work.
Kind regards
Round 2
Reviewer 1 Report
Comments and Suggestions for Authors
The authors have done a great job revising their manuscript.
Author Response
We would like to express our gratitude for your positive feedback and appreciation of our revised manuscript. We are pleased that the revisions have met your expectations, and we believe that the changes have strengthened the quality of the paper.